# What Is Truly Informed Consent in Medical Practice and What Has the Perception of Risk Got to Do with It?

**DOI:** 10.3390/healthcare13010008

**Published:** 2024-12-24

**Authors:** Catherine Jane Calderwood, Geir Sverre Braut, Siri Wiig

**Affiliations:** 1Faculty of Science, University of Strathclyde, Glasgow G1 1XQ, UK; catherine.calderwood@strath.ac.uk; 2Stavanger University Hospital, N-4068 Stavanger, Norway; geir.sverre.braut@sus.no; 3SHARE—Centre for Resilience in Healthcare, Faculty of Health Sciences, University of Stavanger, N-4036 Stavanger, Norway

**Keywords:** consent, patient safety culture, risk, risk perception, compensation

## Abstract

Making decisions about risk, describing and appropriately explaining risk in medical practice is complex for patients and professionals. In this paper, we investigate how the concept of consent is practiced differently in the UK and Norway and discuss pros and cons of the chosen approaches from a patient safety culture perspective. We argue that consent is a fundamental part of the safety culture and influence on health system functioning and patient and staff safety. Examples from the UK and Norway are used and discussed in terms of how risk perception influences consent processes and practices.

## 1. Introduction

Risk is fundamentally complex in medical practice, and medical practice involves the risk of patient harm and complications. Making decisions about risk, describing and appropriately explaining risk are more complex still—for patients and professionals. The perception of risk has many facets [1]. Professionals are expected to be up to date with the latest research, facts and figures, guidelines, and rapidly changing information whilst working in a highly pressured clinical role. For the patient, their perception of risk, preferences, and willingness to accept risk can be based on their own values, their past experience, the health dilemma presented, and how it is impacting on their life, their understanding, and interest, whether the stakes are large or small. How should knowledge about risk be communicated to the patient? How should patient preferences be considered and integrated in shared decision-making processes? What level of risk is acceptable not to explain? How much does fear of litigation and a culture of blame drive the process, and do different legal systems influence clinical practice? To exemplify, Table 1 shows the diversity in understanding and talking about risk and how this may play out in practice based on a real-life clinical situation.

As exemplified above, there are numerous layers when we talk about risk and risk perception and how people, patients, and professionals perceive, balance, and weight different information [1]. They relate to individual needs, competences, and preferences; they relate to cultural traits and attitudes of how patient safety is conceptualised in a system, unit, or in terms of individual responsibility and who is responsible, accountable, and to blame; they relate to the role of the treatment indication for professionals compared to decision-making for the patient concerning the available treatment, alternatives, and evidence; and they relate to contextual settings and regulatory frameworks in different healthcare systems. In health care, making decisions about risk and if and how to intervene happen in the relationship between the professionals and the patients and happen many times on a daily basis. This implies informing and consenting as a core element in health care [2,3]. However, we may question whether consenting is core in the patient safety culture debate and whether it should be? Or does this field focus more on structural and cultural elements in how systems are set up to prevent and respond to adverse events. However, when poor outcomes occur, one will turn to medical records to see what happened during the consultation and treatment process, and how well patients were informed about the risks, possible benefits, likelihood, and consequences, whether they consented based on sufficient information, and also whether they had the capacity to consent. There is not necessarily a link between patient safety measures, informed consent, and the patient safety culture, but this is a highly dynamic field with a continuous need for an ongoing debate to keep up with new developments, risks, and demands from different parties.

In this paper we aim to pick up this debate and draw on how the concept of consent is practiced differently in the UK and Norway and discuss pros and cons of the chosen approaches from a patient safety culture perspective. We argue that consent is a fundamental part of the safety culture and influence on health system functioning and patient and staff safety. This will be illustrated based on examples from the UK and Norway, and discussed in terms of how risk perception could influence this and how compensation schemes are set up and could potentially play a role in medical decision-making either in a defensive or risk-seeking direction. Risk perception may be amplified or attenuated based on socio-cultural elements and structures in any healthcare system [1] and should be placed higher on the agenda when we try to understand the role of consent and how it relates to safety culture. Diverse risk perception between patients and professionals may exist, but the ability to influence decisions may vary between professionals and across cultural settings and norms.

## 2. Informed Consent—What Is It?

By ‘informed consent’, we mean the process of providing patients with sufficient information to enable them to make a voluntary and informed decision regarding whether to undergo a procedure or not, and that the patients are able to understand the information given [4]. Consent from the patient must be obtained for all interactions, e.g., an examination, a blood test, an onward referral. This may be oral, written, or implied—such as when a patient offers their arm for a requested blood test. European countries have differing expectations and guidance when it comes to informed consent for operative or invasive procedures—some require written informed consent (e.g., the UK), while others inform but do not obtain patients’ consent in a written format (e.g., Norway). Does it matter?

The literature shows that from the patients’ perspective, consent forms may fall short of providing information to inform decision-making. They may be too complex and difficult to read and comprehend; hence, the patient–doctor conversation is recognised for its importance [5]. From a medical point of view, how doctors inform and ensure patients are provided with sufficient information and making sure they understand it is a fundamental part of medical practice. This relates to how they are culturally and socially influenced in their risk understanding, or risk-seeking or risk-averse behaviour in decision-making processes. The need for balancing information provision, tailoring information, and assessing patients’ competence and literacy level, with an increasingly more up-to-date and healthcare-literate population with access to medical information online may challenge this information and consent process for clinicians.

## 3. Principles and Practice for Informed Consent in the UK and Norway

In the Norwegian legislation on patient’s rights, it is stated that health care, as a rule, shall be based upon a “valid consent” from the person concerned, as stated in Section 4.1 in the Patients’ Rights Act. The concept of valid consent should be interpreted in a wider context than “informed consent” [6]. In addition to sufficient information being given, the patient must be cognitively capable of understanding the implications and there must be real freedom to refuse accepting the care offered. This puts an obligation upon health professionals to ensure that the patient has a minimum understanding of the risk following the suggested intervention and the risk related to not accepting the offered services. The Act also states that if the patient withdraws their consent, information shall be given on the possible consequences of such a decision.

There are only a few situations where written consent is practiced and required by law. These are prenatal diagnostics, sterilisation, altruistic organ donation, genetic therapies, and assisted fertilisation, chosen in particular because they are not deemed an ‘essential medical intervention or care’. In all other cases, conversation and discussion regarding surgery and intervention is recorded in the case notes according to the legal requirements but neither the patient nor the practitioner sign a document.

In the UK, guidance for shared decision-making and informed consent was updated by the General Medical Council in November 2020 [7]. This followed the high-profile case of a pregnant woman, Nadine Montgomery, who had diabetes. Due to complications during his birth more than 20 years ago, her son has significant disability and lifelong care needs. In the Montgomery court case in 2015 [8], the court ruled that if Ms Montgomery had been made aware of all the possible risks of a vaginal birth, she would have chosen a caesarean section. For the first time, this put the patient rather than the professional in the ‘driving seat’ regarding how much information and detail of risk should be given and how informed consent should be obtained. Table 2 [7] (p. 5) illustrates what are considered the seven principles for shared decision-making and informed consent from this document.

In points 6 and 7 above, capacity is described as decision-specific and time-specific; so, a person can only have capacity or lack capacity to make a specific decision at a specific time. Each jurisdiction of the UK has its own mental capacity legislation which, together with accompanying codes of practice, provides a framework for making decisions when patients lack the capacity to decide for themselves.

In the UK, the legal need for informed consent is described in the Health and Social Care Act 2008 Regulations. It is standard procedure for surgical operations and invasive procedures and interventions to require written consent from the patient. However, the GMC guidance states: ‘Obtaining a patient’s consent needn’t always be a formal, time-consuming process. While some interventions require a patient’s signature on a form, for most healthcare decisions, you can rely on a patient’s verbal consent, as long as you are satisfied they’ve had the opportunity to consider any relevant information and decided to go ahead. Although a patient can give consent verbally (or non-verbally) you should make sure this is recorded in their notes [7] (p. 9)’.

Despite the rhetoric that the signature is hugely significant, the reality is that consent should be a process with two-way dialogue, as above, and many UK doctors would also be surprised at the actual GMC wording regarding the consent form [7]. ‘Consent forms can be a helpful prompt to share key information, as well as a standard way to record a decision that can make regular review easier. Consent forms can also be used to review decisions made at an earlier stage, and the relevant information they were based on. But, filling in a consent form isn’t a substitute for a meaningful dialogue tailored to the individual patient’s needs’.

The use of standardised consent forms with pre-printed information to reduce variation and increase accuracy has followed on from the Montgomery case. Giving people the opportunity to consider their options over a period of time, with information to take home to discuss with family and their General Practitioner (GP), is sometimes offered. Generic forms are provided to try to ensure all aspects of significant risk are disclosed and this is obviously dependent on the nature of the procedure being proposed. The benefits, alternatives, and unexpected procedures that may be required during the operation are flagged and patient opinion is sought as well as highlighting the option of doing nothing. A recent study has shown the improved accuracy and reduced omission error rate when consent is obtained using a digital form, and importantly, this also improved the patient experience of shared decision-making [9]. Practitioners believe this ensures fully informed consent, protecting both the patient and doctor. This practice is not universal, although it is increasing. Inevitably, differences in knowledge, experience, time constraints, complexity level, and individual patient characteristics will lead to variation. Of course, as most procedures end without harm, these processes are not further examined in the vast majority of cases, in contrast to when there is an adverse outcome. Also worth noting is when harm occurs, a lack of information and lack of communication may add a burden to the problematic situation for the patient and their families. 

As described, the Norwegian and UK practices are different. However, the requirements for informing patients about the risk associated with the relevant procedure and involving patients in the decision-making process is equally as important in both countries.

## 4. Compensation Schemes from Medical Harm

Routes for compensation are very different in the two countries. Norwegian legislation clearly defines right and duties for informing and making decisions with patients. The Norwegian System of Patient Injury Compensation (NPE)—a government-funded compensation scheme—allows complaints to be brought forward by the patient in an application for compensation after possibly being harmed. In cases where patients experience financial loss as a result of an injury caused by insufficient medical treatment, compensation will be approved if the application fulfils specific conditions [10]. As a patient, you are, in principle, entitled to receive economic compensation if you suffer a harm caused by a failure in the provision of health care, even if nobody is to blame, and you have or probably will experience economic losses due to this harm. The complaint is, in the first instance, considered by legal and medical officers in the NPE, who decide whether there has been substandard care and award a level of compensation that is determined by the level of harm caused and the long-term needs of the patient. The decision made by the NPE can be brought forward to an expert panel for review and a final decision if the patient is not content with the decision in the first instance.

This compensation scheme is funded by public taxes and the handling of cases is accessible at no cost to all of the Norwegian population. The providers of health care in Norway, public as well as private, are obliged by law to take part in this system, and according to specific regulations, also have to pay an annual fee for participation. Every provider of health care is also obliged to give information to the NPE so that the cases may be investigated. Apart from this, the care providers, either individuals or institutions, have no other obligations, e.g., related to meeting with a panel or being available for oral questioning. The legal parts in this system are merely the “public” represented by the NPE and the single patient. Information given to the NPE will not usually be used in the investigation of the practice of individual healthcare professionals, neither by the supervisory board nor the police.

In the UK, there is no such overall scheme at no cost to patients. Individual patients can, through the legal system, make a claim for compensation if harm has occurred and can be proven. Legal and other costs are incurred by the individual and may be recouped if the case is successful. Compensation is paid from NHS resources. Patients may receive compensation for injury, and also for loss of work and future earnings, and for costs of long-term care. Professionals have indemnity provided via their NHS employer. The Montgomery case marked a shift away from the previous legal test of duty of care, which was that the healthcare practitioner must have acted in a way that fell short of acceptable professional standards. Known as the ‘Bolam’ principle, this tested whether the actions of the health professional in question could be supported by a ‘responsible body of clinical opinion’. Thus, the individual’s practice was examined to apportion blame and prove negligence regardless of systemic failures. Montgomery has changed the context for consent from the position of the clinician to that of the patient.

## 5. Discussion

How do we achieve truly informed consent in a fast-paced system with demands on time and resources, and should or does legal practice have influence? What is certain is that optimizing shared decision-making and truly informed consent is best practice and should be at the core of healthcare interactions. Up until now, conversations about risk and options regarding procedures have been in the domain of individual practitioners with a relationship with their patients. Recent progress has been made towards standardizing information given to patients and to ensuring that this is timely, at an appropriate level, and that patients and their families understand the options. It is clear than even in outwardly similar European countries, the compensation schemes for harm in health care are markedly different, and even what would appear to be the simplest of interventions—signing or not signing a consent form—are not acknowledged as necessary in both systems. The responsibility of an individual working in a complex healthcare system also opens up a very interesting debate. Compensation schemes have evolved differently and several proposals for ‘no blame’ schemes in the UK have been considered and debated, so far unsuccessfully, in recent times.

Compensation schemes are not necessarily connected with the principles and practice related to informed consent. However, a system requiring evidence of guilt as a prerequisite for compensation will invite to establishing formalised and strict procedures to document the agreed terms and information provided between the patient and the care provider.

The major difference between Norway and the UK from a patient safety culture perspective relates to the UK mandatory written consent and compensation claims handled in the legal system when fault or negligence by an individual is often found; while Norway has established a compensation system outside if the legal court system, individual practitioners are not held to account with regard to financial compensation, and in most cases do not practice obtaining written consent. This does not mean one of the systems is better than the other, or that some patients are better informed with either approach. However, we argue this needs more attention as resources will most likely be more constrained in the future, patients will be more informed, and technological solutions and artificial intelligence will contribute to pushing the boundaries of knowledge and medical treatment options. Still, doctors and patients are the ones who need to make decisions and consent to treatment and care based on the available information.

## 6. Conclusions

What, therefore, is truly informed consent? As discussed, the challenges of understanding risk, describing risk, managing uncertainty and personal values, and views and preferences (variable between both patients and professionals, of course) are complex and layered [11]. In the UK, the need for written consent is seen as a safeguard for both parties, but in reality, the requirements for informed consent are the same in both countries. The patient should be enabled to understand the risks involved in health care, source information relevant to them, and accept treatment with as much appropriate knowledge as possible. This also includes patients with low health literacy, resources, and abilities to communicate. In ‘safety’ terms, the fundamentals of good communication, respect for the individual, and involvement in decisions are a necessity. The ability of the patient to speak up if these fundamentals are not met for them is fundamental, but often proves difficult in practice for many practical reasons. Until now, consent has not, we believe, been part of the patient safety ‘culture’, integrated in training and teaching and good work practice and discussed in the literature. During the scrutiny that comes after an adverse event, the fundamental principles of informed decision-making and consent are carefully examined. However, we may question whether this aspect of safety, which is of course the only aspect the patient has any control over, has received the due diligence and input it rightly deserves. Further attention is needed to enable continuous reflection for professionals and regulators. The field is changing rapidly as access to information and technological advancement will play a key role in future healthcare provision and decision-making

## 7. Future Directions

We would all want to be fully informed in order to understand the risks and options we may have as patients, and also the possible decision to say no and do nothing. This is an ongoing debate in many counties [12,13] and if 10% of all health care causes harm [14], comprehensive information should be front and centre of how the present-day safety culture matures. This may lead to patients deciding not to have any treatment at all. The elements of risk, perception, uncertainty, and consent should be more strongly integrated in the future patient safety culture debate to enable sound discussions and decision-making in healthcare provision.

## Figures and Tables

**Table 1 healthcare-13-00008-t001:** Clinical example of risk-based decisions.

Making Decisions Based on Risk in Maternity Services
As a practising obstetrician, first author CC was once counselling 2 women in succession who had screening for genetic abnormalities in early pregnancy. The arbitrary cut-off for informing women of a ‘high-risk’ result is that the risk of the baby being affected is greater than 1 in 150. This cut-off was simply determined statistically as encompassing 5% of women who were screened. No other scientific, clinical, moral, ethical, or other considerations are involved. All other women were informed that they had a ‘low-risk’ result. The first woman had not been notified but had requested to see her results. These showed a risk of an affected baby of 1 in 400. She proceeded to a definite diagnostic test (which itself carried a risk of an adverse outcome of 0.5%) as she did not want to tolerate any uncertainty. The next woman had a ‘high-risk’ result of 1 in 10. This level of risk is very unusual, and CC assumed (as with most women with a high-risk result) that she would proceed to a definitive test. However, she asked ‘but doctor this means there’s a 90% chance the baby will be ok?’. Yes was replied and she left the room without further intervention.

**Table 2 healthcare-13-00008-t002:** Principles for informed consent [7] (p. 5).

Principle	Description
1	All patients have the right to be involved in decisions about their treatment and care and be supported to make informed decisions if they are able.
2	Decision-making is an ongoing process focussed on meaningful dialogue: the exchange of relevant information specific to the individual patient.
3	All patients have the right to be listened to, and to be given the information they need to make a decision and the time and support they need to understand it.
4	Doctors must try to find out what matters to patients so they can share relevant information about the benefits and harms of proposed options and reasonable alternatives, including the option to take no action.
5	Doctors must start from the presumption that all adult patients have capacity to make decisions about their treatment and care. A patient can only be judged to lack capacity to make a decision at a specific time, and only after assessment in line with legal requirements.
6	The choice of treatment or care for patients who lack capacity must be of overall benefit to them, and decisions should be made in consultation with those close to them or advocating for them.
7	Patients whose right to consent is affected by law should be supported to be involved in the decision-making process, and to exercise choice if possible.

## Data Availability

Not relevant.

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
