# Peer review of "What Is Truly Informed Consent in Medical Practice and What Has the Perception of Risk Got to Do with It?"

_healthcare, 2024, doi:10.3390/healthcare13010008_

Round 1
Reviewer 1 Report (Previous Reviewer 2)
Comments and Suggestions for Authors
the authors have revised along the suggestions provided, nothing to add
Author Response
Please see the attachment.

Reviewer 2 Report (Previous Reviewer 3)
Comments and Suggestions for Authors
Many thanks for the possibility to read this article, as the issue is pivotal as far as critical comparative discussion. The text is much improved.
I recommend to reference Health and Social Care Act 2008 Regulations.
I suggest also 10.1016/j.jflm.2024.102674
section Compensation schemes from medical harm. Please link the role of informed consent disclosure/absence/issues and risks' disclosure compared to the medical harm. (for instance if there was a patient dissent or the patient was not informed on those complications or risks)
many thanks
Author Response
Please see the attachment.

Reviewer 3 Report (Previous Reviewer 4)
Comments and Suggestions for Authors
Dear Authors.
I believe that the authors induce a link between the informed consent process and patient safety culture when, for me, they are independent. Performing a good informed consent process does not guarantee that all necessary patient safety measures will be taken. On the contrary, all patient safety measures can be taken without having performed an adequate informed consent process.
I believe that the authors also induce the authors to link the informed consent process with compensation for damages, thus distorting the true purpose of this process. The objective of the informed consent process is not the legal protection of the professional, but to allow the free and voluntary participation of patients in everything related to their health in order to guarantee their autonomy.
I believe that the authors continue to offer a well-known and generic view of the informed consent process and informed consent forms.
Kind regards.
Round 2
Reviewer 3 Report (Previous Reviewer 4)
Comments and Suggestions for Authors
Dear Authors.
I have no additional comments to make.
Kind regards.
This manuscript is a resubmission of an earlier submission. The following is a list of the peer review reports and author responses from that submission.
Round 1
Reviewer 1 Report
Comments and Suggestions for Authors
I did not find an 'Opinion' category for work in the template. Please confirm with the editor whether manuscripts of this type are acceptable for publication.
Comments on the Quality of English Language
Please check the grammar of the title.
Reviewer 2 Report
Comments and Suggestions for Authors
I have read this opinion paper with great interest, and do value the paper as submitted, with focus on socio-cultural aspects 'affecting' clinical practices. One may hereby even question the causality (and the direction) of the different 'handling/legal' systems, as the out of court approach in Norway is perhaps also 'just' a reflection of these socio-cultural differences ?
I only have some specific reflections or suggestions for your consideration.
I do miss sufficient reflection on patient preferences, like eg; surgery or medical care for osteoarthosis, deprescribing practices (quite commonly implemented in geriatric care). To concept of shared decision making can perhaps be stressed somewhat clearer ? In the current version, there is quite some focus on safety and avoiding errors, but there is also a positive side of patient involvement and shared decision making (even if workflow or procedures differ).
Second, the title is not so clear that this focus on 'normal clinical care', as consent has a quite different construct and approach in clinical trials. I would recommend to make this clearer in the title, and perhaps also in the abstract.
Minor comments
'highly pressured clinical role', what do you mean with this ?
table 2, principle 7: typo, eb should read be
Line 129, perhaps rights ?
Reviewer 3 Report
Comments and Suggestions for Authors
see word file

Reviewer 4 Report
Comments and Suggestions for Authors
In my opinion, the authors give a well-known and generic view of the informed consent process and informed consent forms.
The purpose of this process is to guarantee the patient's autonomy in all decisions that affect his or her health and not his or her safety or the reparation of harm.
Knowledge of the risks of a procedure is intended to help the patient to make a free decision based on his or her preferences and needs. I consider that their knowledge is not directly related to the patient's safety because all the necessary precautions can be taken to prevent the patient from suffering any harm even if the risks are not explained to the patient.
Kind regards.
